# Mechanical Properties and Very High Cycle Fatigue Behavior of Peak-Aged AA7021 Alloy

**Byung-Hoon Lee [1,2], Sung-Woo Park [1,2], Soong-Keun Hyun [2], In-Sik Cho [3] and Kyung-Taek Kim [1,\*]**

[1]  Advanced Process and Materials R&D Group, Korea Institute of Industrial Technology, 156, Gaetbeol-ro, Yeonsu-gu, Incheon 21999, Korea; byung812@kitech.re.kr (B.-H.L.), parknoel90@kitech.re.kr (S.-W.P.)
[2]  Department of Advanced Science of Engineering, Inha University, 100, Inha-ro, Michuhol-gu, Incheon 22201, Korea; skhyun@inha.ac.kr
[3]  Mbrosia. Co., 70, Sunmoon-ro 22beon-gil, Tangjeong-myeon, Asan-si, Chungcheongnam-do 31460, Korea; mbrosia1018@naver.com
\*  Correspondence: kkt@kitech.re.kr; Tel.: +82-32-850-0213

**Abstract:** The effect of heat treatment condition on non-Cu AA7021 alloy was investigated with respect to mechanical properties and very high cycle fatigue behavior. With a focus on the influence of heat treatment, AA7021 alloy was solution heat-treated at 470 °C for 4 h and aged at 124 °C. Comparing the results of solution-treated and peak-aged AA7021 alloy shows a significant increase in Vickers hardness and tensile strength. The hardness of AA7021 alloy was increased by 65% after aging treatment, and both tensile strength and yield strength were increased by 50~80 MPa in each case. In particular, this paper investigated the very high cycle fatigue behavior of AA7021 alloy with the ultrasonic fatigue testing method using a resonance frequency of 20 kHz. The fatigue results showed that the stress amplitude of peak-aged AA7021 alloy was about 50 MPa higher than the solution-treated alloy at the same fatigue cycles. Furthermore, it was confirmed that the size of the crack initiation site was larger after peak aging than after solution treatment.

**Keywords:** AA7021 alloy; age hardening; hardness; tensile test; high cycle fatigue test

## 1. Introduction

7xxx series aluminum alloys are typically classified as heat-treatable aluminum alloys, and have the highest strength among aluminum alloys used as structural materials. The 7xxx series aluminum alloy of Al-Zn-Mg system has a mechanism in which Mg-Zn precipitates in the GP zone during heat treatment, resulting in an increase in hardness and strength [1]. A typical high-strength aluminum alloy often used in industrial environments is AA7075 alloy; however, when properties such as extrusion workability, weldability, and corrosion resistance are required at the same time, the non-Cu-containing AA7N01 alloy is used [2–4]. Such AA7xxx alloys with superior properties have been used in structural materials, aircraft, automobile parts, and are now also used in portable electronic devices [5]. In particular, AA7021 alloy is designed for the purpose of improving high-temperature formability, and it is possible to control the degree of dynamic recrystallization during extrusion by controlling additive elements [6].

Much recent research has been conducted on controlling the microstructures and additive elements in order to manufacture heat-treatable high-strength aluminum alloys [7,8]. In the past, 6xxx series aluminum alloys were classified as light-weight alloys; therefore, much research was carried out with the aim of replacing steel in automobile industries. Recently, studies on the improvement of workability of 7xxx series alloys with superior mechanical properties have been actively carried out to

replace 6xxx series alloys [5,6]. However, the results of the evaluation of the basic properties of the alloys selected in this paper are still insufficient. In particular, there is a lack of data and studies on dynamic modulus of elasticity and high cycle fatigue properties of AA7021 alloy.

Therefore, in this study, we attempted to obtain optimal heat treatment conditions by establishing the criteria for peak aging and over aging conditions through the evaluation of aging time after solution heat treatment of AA7021 alloy. Moreover, tensile testing and very high cycle fatigue testing were conducted to evaluate tensile properties, strain energy densities and high-cycle fatigue behaviors.

## 2. Experimental Procedure

In this study, a non-Cu AA7021 aluminum alloy with Al-5.8Zn-1.6Mg composition was fabricated by continuous casting. The heat treatment of AA7021 alloy was analyzed by JMatPro (JMatPro 7.0, Solution Lab, Daejeon, Korea) simulation and the phases expected to precipitate during heat treatment were analyzed. The heat treatment was carried out in the T6 condition and in the atmosphere, and the solution treatment was maintained at 470 °C for 4 h and aged at 124 °C.

To examine the hardness of AA7021 alloy after aging treatment, the specimens were prepared at a size of $10 \times 10 \times 10$ mm$^3$ between the outer and the center of the billet, and the hardness for each specimen was measured by a Vickers Hardness Tester (HMV-2T E, Shimadzu, Kyoto, Japan). The micro-hardness was measured under the condition of a 200 g load of indicator for 10 s per test, and the average hardness was calculated by measuring 20 times per specimen.

To observe the change of microstructure due to the heat treatment, specimens were prepared under the same conditions as the hardness test. The specimens were polished with #2000 emery paper, mirror-polished with 1 μm diamond suspension, and etched with Modified Keller's Reagent (175 mL $H_2O$, 20 mL $HNO_3$, 3 mL HCl, 2 mL HF), and then observed under an optical microscope.

Tensile tests were carried out at a rate of 1 mm/min using a Mechanical Universal Testing Machine (AG-300knX, Shimadzu, Kyoto, Japan) to confirm the change in tensile properties of AA7021 alloy with heat treatment and aging time. Specimens for the tensile test were prepared according to the ASTM E8 standard, which is shown in Figure 1a. To confirm the tensile properties due to the heat treatment effect, tests were performed by selecting as-cast, solution-treated, peak-aged and over-aged specimens. The tensile strength, yield strength, strain energy density and modulus of resilience of the specimens obtained from the tensile test results are compared, and the stress–strain relationship is shown in the figure.

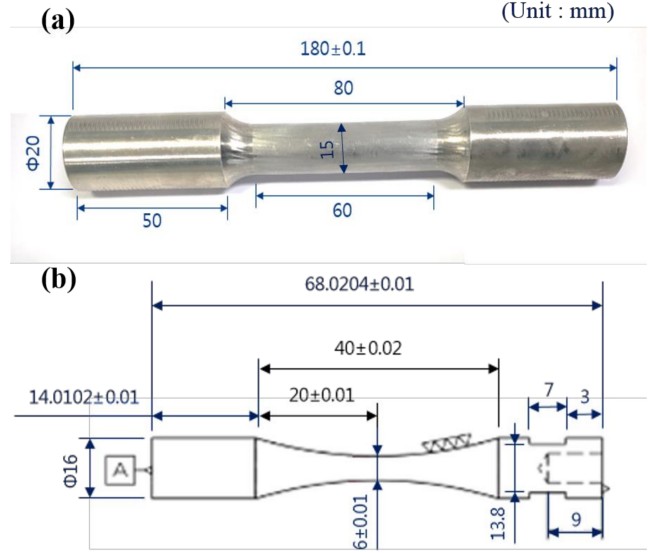

**Figure 1.** Drawing of tensile specimen (**a**) and fatigue specimen (**b**).

The fatigue behaviors of AA7021 alloy were evaluated by the resonance testing method using a 20 kHz elastic vibration wave from an ultrasonic fatigue tester within the range from $10^6$ to $10^9$ cycles. The specimens for ultrasonic fatigue testing were fabricated to match a resonance frequency of 20 kHz, as shown in Figure 1b, based on the dynamic modulus of elasticity of the material measured by IET (Impulse Excitation Technique) method [9,10]. Fatigue life evaluation was performed under two conditions of solution treatment and peak aging of AA7021 alloy. The fatigue test was performed at a resonance frequency of 20 kHz, a stress ratio of −1, and a high cycle area of $10^6$ to $10^9$ cycles.

## 3. Results

### 3.1. Optimization of Heat Treatment Condition of AA7021

Figure 2 is a graph showing the change in hardness of the AA7021 alloy with aging treatment time. The average hardness of AA7021 alloy before heat treatment was 105 HV, and this decreased to 98 HV after solution treatment. The average hardness of AA7021 gradually increased after aging treatment, and it showed the highest value of 148 HV after 32 h of aging. Thereafter, the hardness of the alloy decreased again, and the average hardness was obtained to be about 125 HV after 48 h of aging treatment. From the result of the hardness measurement, the aging of 32 h with the highest hardness was set to the peak aging condition, and the aging of 48 h, which indicated a decrease in hardness, was set to the over-aging condition.

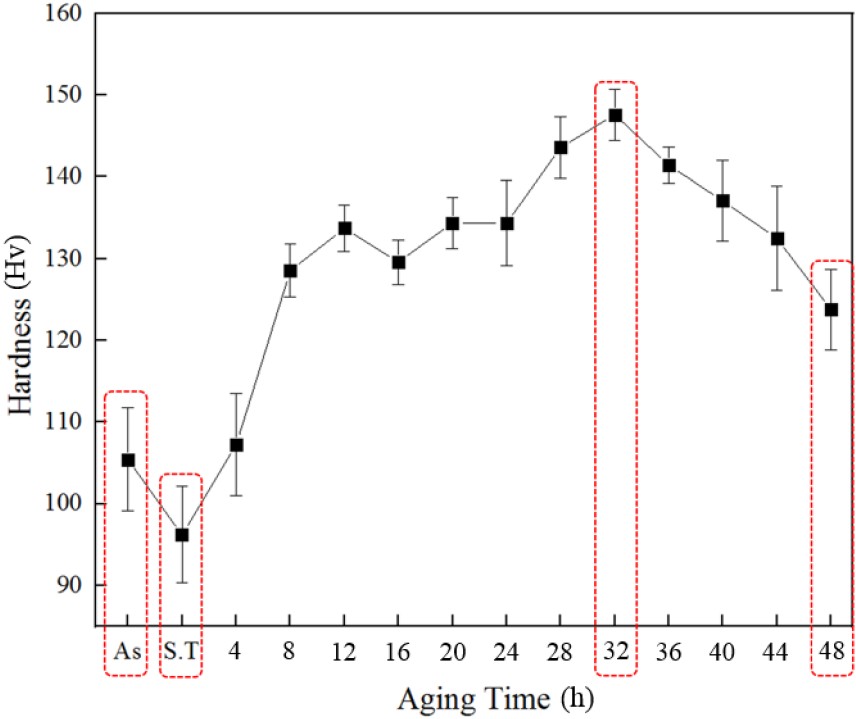

**Figure 2.** Vickers hardness changes of AA7021 with aging treatment time.

### 3.2. Optical Micrographs of AA7021 Alloy

Figure 3 shows the optical microstructures of as-cast, solution-treated, peak-aged and over-aged AA7021. The grain size of the alloy according to the heat treatment condition was calculated by the Line Intercept method. The result of calculation showed that the average grain size of the AA7021 alloy before heat treatment was about 55 μm. However, the grain size of the alloy decreased to about 30–35 μm after peak aging and over aging treatment. These results indicate that the heat treatment affects the grain size of the alloy and increases the uniformity of the structure.

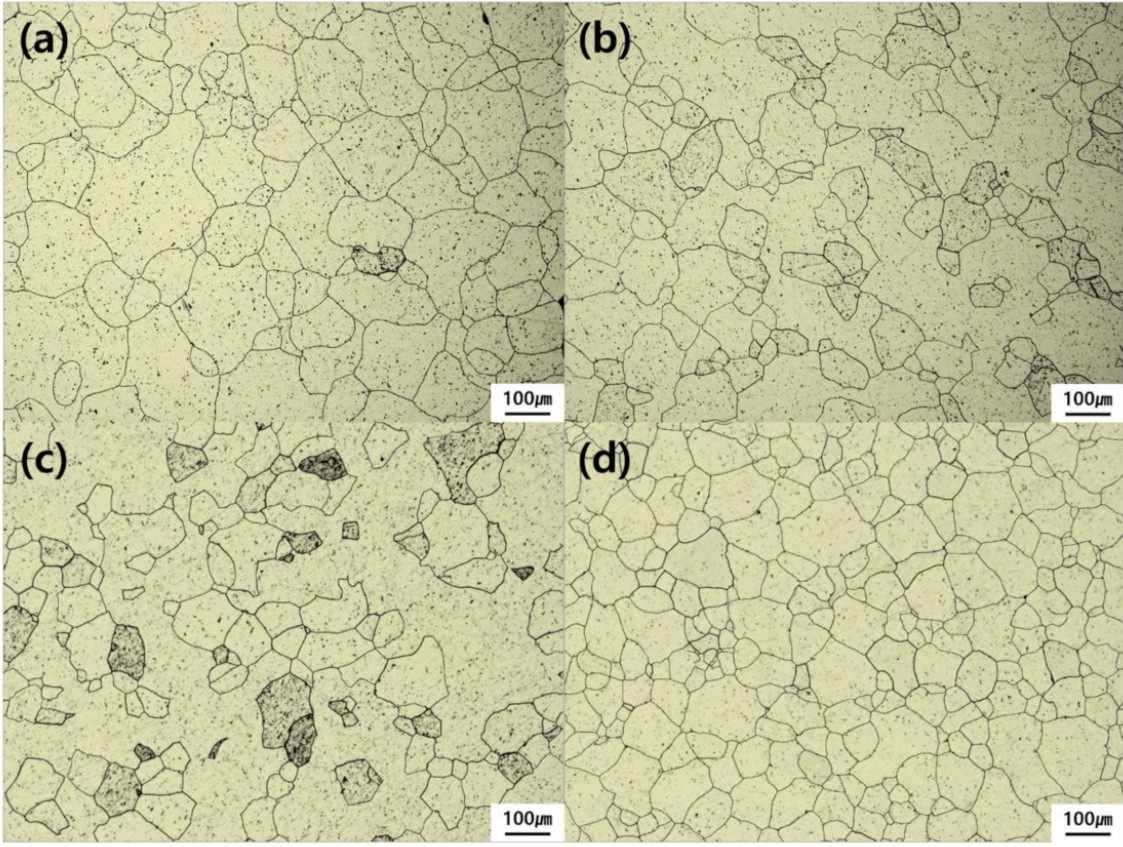

**Figure 3.** Optical micrographs of the as-cast (**a**), solution-treated (**b**), peak-aged (**c**), and over-aged (**d**) AA7021 alloy.

### 3.3. Phase Analysis

X-Ray Diffraction analysis was carried out to analyze the precipitate phases of AA7021 according to heat treatment conditions. Figure 4 shows the results of XRD analysis of as-cast, solution-treated, peak-aged and over-aged specimens of AA7021. The XRD analysis results show that Mg-Zn intermediate phases, which are representative precipitate phases of the 7xxx series, were supersaturated into matrix after solution treatment, compared with the 'as-cast' results. Mg-Zn precipitates can be identified at 38° and 43° peaks in peak aging treatment, and therefore, the precipitation hardening effect can be expected by aging treatment.

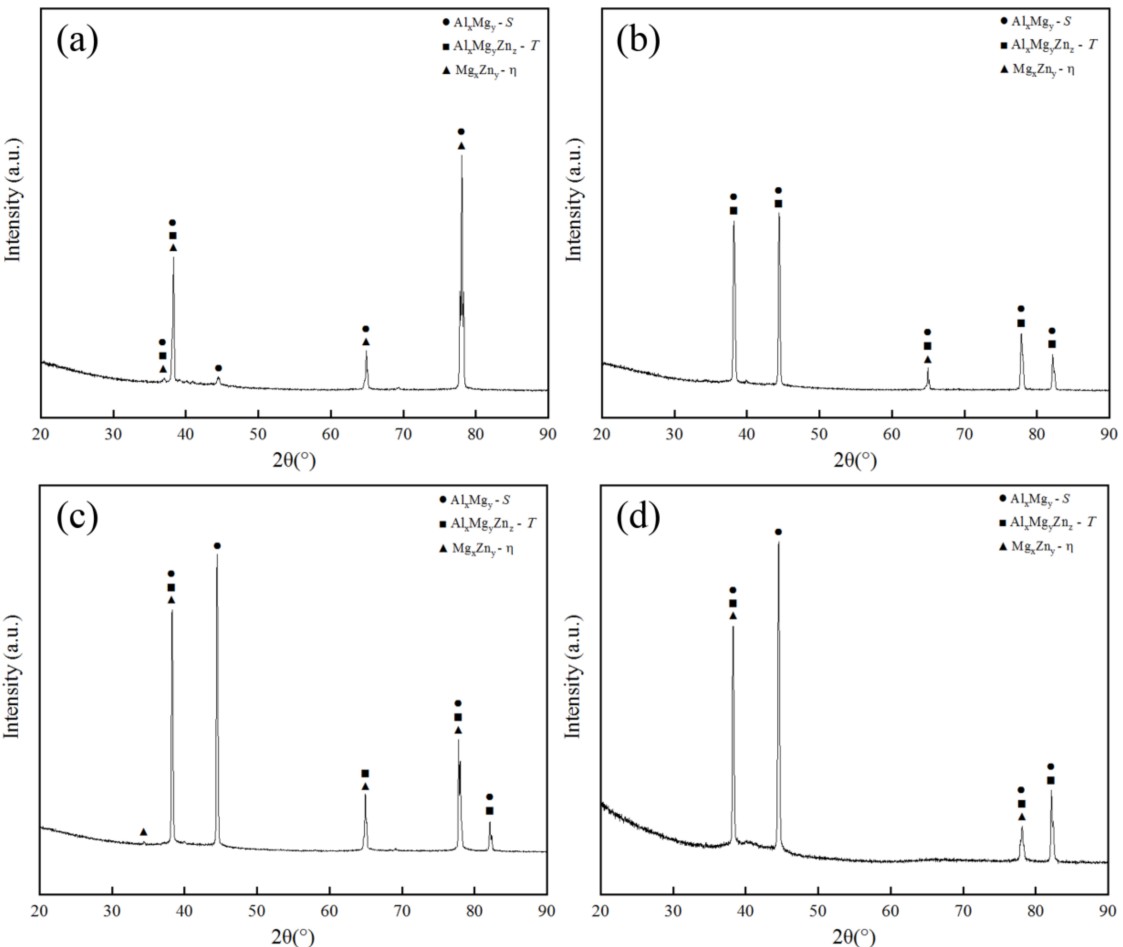

**Figure 4.** Phase analysis by X-Ray Diffraction results of the as-cast (**a**), solution-treated (**b**), peak-aged (**c**), and over-aged (**d**) AA7021 alloy.

### 3.4. Tensile Test of AA7021 Alloy

Figure 5 shows the tensile test results of the AA7021 alloy according to the heat treatment conditions as a stress–strain graph, and the results are organized in Table 1. From the results of the tensile test, AA7021 alloy before heat treatment showed the UTS of 350 MPa, the yield strength of 245 MPa and the 6% of elongation. After solution treatment, the UTS increased to 375 MPa, and the elongation significantly increased up to 18%. On the other hand, the yield strength decreased to 200 MPa. The modulus of elasticity is related to the atomic bonding of the material and there was no significant change in the modulus of elasticity due to heat treatment, so it was judged that there was no effect between those heat treatment process and atomic bonding. After aging treatment, the UTS significantly increased to 440 MPa, and the yield strength increased to 300 MPa. Unlike hardness results, UTS and yield strength increased after over aging, but the difference was minimal. On the contrary, the peak-aged specimen showed 3% higher elongation compared to the over-aged one. The strain energy density and elastic modulus will be discussed in the discussion section, and consequently, the strain energy density follows the tendency of elongation and the modulus of resilience follows that of yield strength.

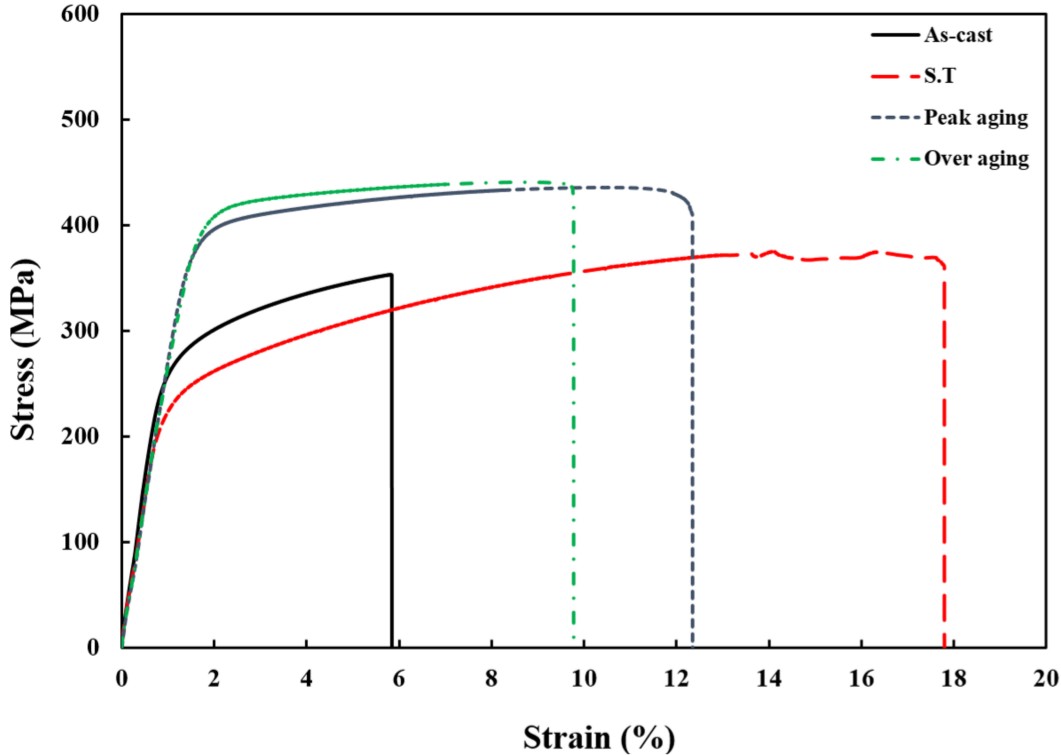

**Figure 5.** Stress–strain graphs according to tensile test of AA7021 alloy.

**Table 1.** Tensile test results of AA7021 alloy by heat treatment condition.

| Heat Treatment | Tensile Strength | Elongation | Modulus of Elasticity | Yield Strength (0.2% Off-Set) | Strain Energy Density | Modulus of Resilience |
|---|---|---|---|---|---|---|
| | MPa | % | GPa | MPa | J/mm³ | J/mm³ |
| As-cast | 352.64 | 5.84 | 69.52 | 244.94 | 17.40 | 1.29 |
| Solution Treatment | 375.18 | 17.81 | 69.67 | 202.14 | 57.93 | 0.86 |
| Peak Aging | 438.75 | 12.37 | 69.25 | 295.41 | 48.93 | 1.65 |
| Over Aging | 440.01 | 9.78 | 69.65 | 310.18 | 38.53 | 1.98 |

*3.5. High-Cycle Fatigue Test of AA7021 Alloy*

Figure 6 shows the S-N curves of the fatigue test results obtained from the solution-treated and the peak-aged AA7021 alloy. Comparing the results between solution-treated and peak-aged fatigue specimens shows that the peak-aged specimen has 50 MPa higher stress amplitude than the solution-treated specimen in the same fatigue cycles. The overall S-N curve trend shows that the peak-aged specimen has a higher fatigue endurance limit than the solution-treated one.

Fatigue fracture analysis of specimens was confirmed by SEM, and the results are shown in Figure 7, which are the SEM images of fatigue fracture, with failure in the ranges of $<10^6$, $10^6$–$10^7$ and upper $10^7$ cycles of the solution-treated and peak-aged AA7021 alloys. Both the solution-treated and peak-aged specimens showed a wider crack initiation site as the fatigue cycle increased [10]. In particular, in the case of peak aging, the crack initiation site occupied more than 1/3 of the total fracture area in the high cycle fatigue range of $10^7$ cycles or more. The area of crack initiation sites, which were more than 10% of the total area, are shown in Figure 6 as dashed marks. The solution-treated specimen had an initiation site of over 10% at above $10^8$ cycles. The peak-aged specimen, on the other hand, had an initiation site of over 10% at $10^7$ cycles, which had a much lower fatigue region than the solution-treated specimen.

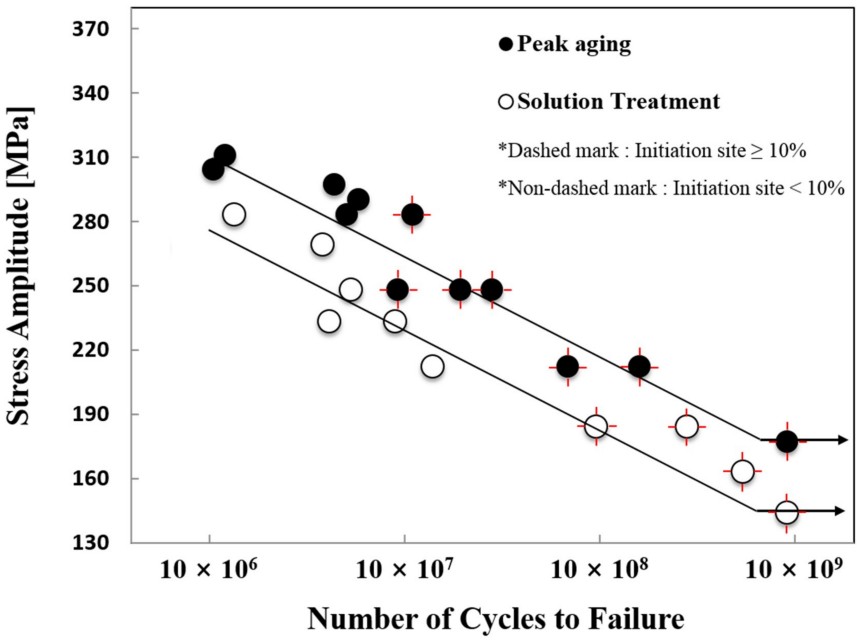

**Figure 6.** S-N curve according to high cycle fatigue of AA7021 alloy.

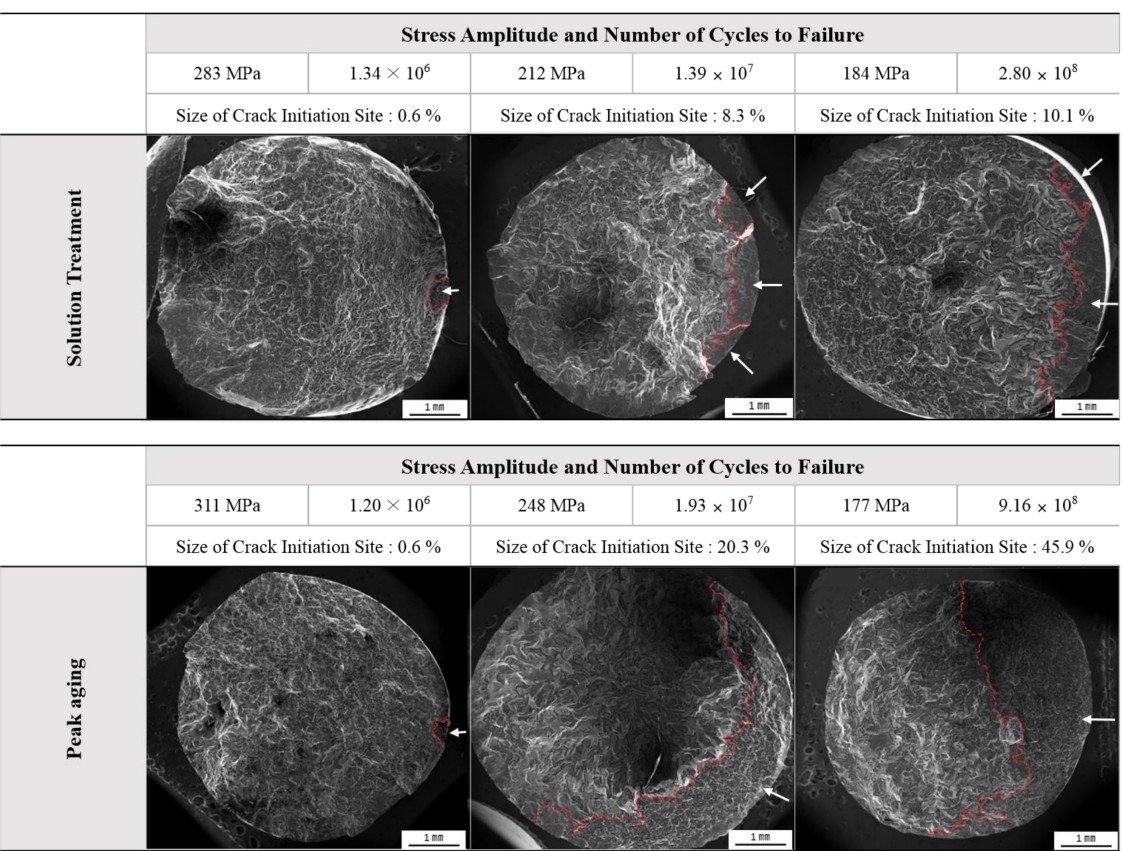

**Figure 7.** Fatigue test results of AA7021 alloy by heat treatment condition.

## 4. Discussion

The effect of heat treatment and aging on AA7021 alloy was analyzed by tensile test and fatigue test. Based on the results of hardness measurement and tensile testing, the hardness and UTS of AA7021 alloy increased after aging treatment. Also, from the microstructure observation results, the

grain size decreased by about 20 μm after aging treatment compared with before the heat treatment. In terms of material engineering, grain boundaries serve as obstacles to slip. Small grain size hinders slip movement and causes matrix strengthening [11]. In addition, the larger the misorientation of grain boundary angles, the larger the obstacle effect becomes. In general, the correlation between the grain size and the strength of the material follows the Hall-Petch equation,

$$\sigma = \sigma_0 + \mathrm{k}d^{-0.5} \tag{1}$$

where *d* is grain size and $\sigma_0$ and k are constants [12]. In the Hall-Petch relation, the grain size and the strength are in inverse proportion; therefore, the strength increases as the grain size decreases.

The precipitation of intermetallic phase by aging treatment was confirmed by XRD analysis. Intermetallic phases, which were precipitated by the particles in the matrix, increased the mechanical properties in particle-reinforced aluminum. Kouzeli and Mortensen [13] used the concept of geometrically necessary dislocations to rationalize this effect of size on the yield behavior of the alloying composition. Dislocations are generated when plastic deformation occurs. Dislocations move until there is nowhere to go (e.g., grain boundaries) and are filed up; this storage causes the material to work-harden. The relationship between dislocation mobility and strengthening mechanisms has been treated by Yonenaga, Motoki [14] and Fleischer [15]. In particular, according to Kouzeli and Mortensen [13], in the case of particle-reinforced metals, the incompatibility in deformation between the plastically deforming matrix and the essentially rigid particles leads to the creation of strong strain gradients in the metallic matrix. Thus, for a given geometry of plastic flow, a finer composite microstructure should lead to a greater strain gradient in the composite matrix, which, in turn, should result in a greater density of geometrically necessary dislocations and a higher composite flow stress. In the case of 7xxx-series aluminum alloys, Mg-Zn precipitates play the role of particles, resulting in a decrease in dislocation mobility. In addition, the greater dispersion of Mg-Zn precipitates leads to a greater effect.

According to the results of the tensile test, the highest strain energy density of AA7021 was shown when the alloy was solution-treated, and was lower when aging treatment became longer. Strain energy density is defined as the energy absorption capacity of the material through plastic deformation until failure occurs, which means the unit area from the stress–strain graph in the tensile test. In other words, high strain energy density means that the material has a high resistance to the load applied from plastic deformation to failure. In the case of AA7021 alloy, the over-aged specimen had moderately higher UTS and yield strength than the peak-aged one, but the peak-aged one had significantly high elongation, which led to the higher strain energy density. In addition, the low elongation and strain energy density of the as-casted AA7021 alloy is judged by the influence of defects and pores occurred during the casting process.

The modulus of resilience refers to the energy absorbing capacity due to elastic deformation and the energy recovery due to load removal, and the unit is the product of the two axes, $J/mm^3$, expressed as the absorbed energy per unit volume of the material. The yield strength and the modulus of resilience are closely related to the fatigue life. High yield strength and modulus of resilience are associated with higher resistance to fatigue stress.

Generally, for high fatigue life, crack initiation is responsible for 90% of total fatigue life, and crack propagation is the factor that determines the fatigue life of the material in low cycle fatigue. Since aluminum alloy has no significant fatigue limit, it is difficult to analyze the fatigue fracture plane. However, as shown in Figure 7, the crack initiation site appears at the small edge of the surface at $10^6$ cycles, i.e., at the high stress amplitude zone, and the crack initiation site becomes wider as it moves to the very high cycle region, i.e., at the lower stress amplitude zone [16–18]. The larger crack initiation site is considered to have a high fatigue life due to stress concentration dispersion until crack initiation site formation, and the results of Figure 6 show that the fatigue life of the peak-aged specimen was higher than that of the solution-treated one. The following conclusions can be deduced from the results of yield strength of the alloy as well as the size of crack initiation site. Moreover, Ahn [19] concluded

that smaller grain size and grain refinement increase the fatigue strength of the alloy by 30%, which is very similar to our results.

## 5. Conclusions

For the purposes of improving the high temperature formability, 7xxx series aluminum alloy of non-Cu element was fabricated in this study. In order carry out detailed characterization of the alloy, optimum heat treatment conditions were established by simulation, and hardness, microstructure, phase analysis, tensile test and fatigue tests were performed for verification, and the results are summarized as follows.

(1)  The hardness after the solution treatment was reduced as compared with as-cast, and the hardness increased with aging treatment. The hardness after aging of 32 h showed the highest value of 148 HV, and it gradually decreased after 32 h of aging. In particular, the grain size during aging treatment was reduced by 20 μm compared to that of the as-cast specimen, and the Mg-Zn intermediate peak was confirmed from XRD.

(2)  The peak-aged specimen showed higher UTS than the as-cast one by 90 MPa, higher yield strength by 50 MPa and the elongation was doubled to 12%.

(3)  The yield strength and the modulus of resilience due to the peak aging treatment were higher than those of the solution-treated specimen, which could be expected to lead to higher fatigue life, as the yield strength and the modulus of resilience are related to the fatigue strength.

(4)  At the high stress amplitude zone, the crack initiation site was visible at the small edge of the surface, and it was widely distributed at the low stress amplitude zone, which is in the very high cycle region.

**Author Contributions:** Conceptualization, Byung-Hoon Lee; Data curation, Byung-Hoon Lee and Sung-Woo Park; Formal analysis, Byung-Hoon Lee; Funding acquisition, Kyung-Taek Kim; Investigation, Byung-Hoon Lee, Sung-Woo Park and In-Sik Cho; Methodology, Byung-Hoon Lee and In-Sik Cho; Project administration, Kyung-Taek Kim; Resources, Soong-Keun Hyun; Software, In-Sik Cho; Supervision, Kyung-Taek Kim; Validation, Soong-Keun Hyun and Kyung-Taek Kim; Visualization, Byung-Hoon Lee and In-Sik Cho; Writing-original draft, Byung-Hoon Lee; Writing-review & editing, In-Sik Cho, Soong-Keun Hyun and Kyung-Taek Kim.

**Funding:** This work was funded by the Korea Institute of Industrial Technology (UR180048) (KITECH, Korea).

**Conflicts of Interest:** The authors declare no conflicts of interest.

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
