# Peer review of "Mechanical Properties and Very High Cycle Fatigue Behavior of Peak-Aged AA7021 Alloy"

_metals, doi:10.3390/met8121023_

Round 1
Reviewer 1 Report
The authors performed an investigation of the strength, hardness, morphology, microstructure and fatigue behaviour of non-Cu Aluminium alloy after heat treatment and aging. They achieved improvement of the properties of the alloy under certain treatment conditions as compared to the untreated or differently treated alloys. It is a systematic investigation, well performed discussed and reported, paper well written. Thus publication of the paper is recommended.
Author Response
Please find the detailed response in the attachment.

Reviewer 2 Report
In this article, the influence of the heat treatment condition on mechanical properties and VHCF on AA7021 alloy is studied.
It is an interesting subject but in the opinion of the reviewer, before assessing its possible publication, it is necessary to clarify how the size of the crack initiation site is defined and measured. Table 2 does not provide information on the location of crack initiation or its size, please clarify this aspect.
It is necessary to improve the appearance of the captions of figure 6.
Author Response

(The authors gave the same response as above.)

Reviewer 3 Report
Prof. Dr. Hugo F. Lopez,
Metals
Manuscript ID: metals-388367
Comments for revision of the paper “Mechanical Properties and Very High Cycle Fatigue Behavior of Peak-Aged AA7021 Alloy”.
General Comments
This paper studies the effect of the alloy AA7021’s thermal treatment on the mechanical properties and fatigue of a high number of cycles .
More recent bibliographic references should be considered.
The results of fatigue should be better discussed.
Figures
The figures must be greatly improved, namely Figures 1, 2, 3, 4, 5, and 6 (improving the quality, image size and font size);
The figures shown in table 2 should clearly be improved (it is not possible to observe the mechanisms of mechanical failure, ...)
1. Pg 2, Ln 55-56
Should replace the phrase "... size of 10 x 10 x 10 mm ..." with "..size of 10 x 10 x 10 mm3 ..".
2. Pg 2, Ln 65-72
The geometry and dimensions of the specimens used in tensile tests must be indicated, as well as figure 5 in the text (which is missing).
3. Pg 4, Ln 94-100
The micrographs relating to figure 3 should be better explained, in particular the effect of the treatments.
4. Pg 6, Ln 127
Explain why the results of the elasticity modulus has 2 decimal digits ±0.2.
5. Pg 7, Ln 145
Does table 2 refer to the fatigue tests (Table 2. Tensile test results of AA7021 alloy by heat treatment condition)?
Author Response

(The authors gave the same response as above.)

Reviewer 4 Report
The manuscript described heat treatment and aging effect on mechanical properties of AA7021 alloy. The manuscript is well written. The reviewer recommends this article to be published after minor revision.
1. In figure 1, the radius of shoulder part and gauge length were not found.
2. Caption of table 2 should be “Fatigue test …”.
Author Response

(The authors gave the same response as above.)

Round 2
Reviewer 2 Report
Please diminish de word size of figure 6 (Number of Cycles and Stress Amplitude).
The paper can be accepted in present form, if the minor correction indicated is done.
Author Response
Dear Editor,
I appreciate for your second review.
I have revised my manuscript as you mentioned on last review, which says
"Please diminish de word size of figure 6 (Number of cycles and stress amplitude)".
To make balance with other figures, the header axis font size decrease from 24 to 18 in figure 6 (Remaining font size decrease from 16 to 14).
In addition, I have revised to improve the quality of figure 1.
Once again, I appreciate for reviewing my manuscript.
Kind regards,
Byung-Hoon Lee
